# Current Perspectives on Idiopathic Intracranial Hypertension without Papilloedema

**DOI:** 10.3390/life11060472

**Published:** 2021-05-24

**Authors:** Susan P. Mollan, Yu Jeat Chong, Olivia Grech, Alex J. Sinclair, Benjamin R. Wakerley

**Affiliations:** 1Birmingham Neuro-Ophthalmology Unit, University Hospitals Birmingham NHS Foundation Trust, Birmingham B15 2TH, UK; yujeat@gmail.com; 2Metabolic Neurology, Institute of Metabolism and Systems Research, University of Birmingham, Edgbaston B15 2TT, UK; oxG958@student.bham.ac.uk (O.G.); a.b.sinclair@bham.ac.uk (A.J.S.); b.wakerley@bham.ac.uk (B.R.W.); 3Centre for Endocrinology, Diabetes and Metabolism, Birmingham Health Partners, Birmingham B15 2TH, UK; 4Department of Neurology, University Hospitals Birmingham NHS Foundation Trust, Birmingham B15 2TH, UK

**Keywords:** diplopia, headache, idiopathic intracranial hypertension without papilloedema, intracranial pressure, lumbar puncture, migraine, optical coherence tomography, pseudotumour cerebri, telemetric monitoring, tinnitus

## Abstract

The pseudotumor cerebri syndrome embraces disorders characterised by raised intracranial pressure, where the commonest symptom is headache (90%). Idiopathic intracranial hypertension without papilloedema (IIHWOP) is increasingly recognised as a source of refractory headache symptoms and resultant neurological disability. Although the majority of patients with IIHWOP are phenotypically similar to those with idiopathic intracranial hypertension (IIH), it remains uncertain as to whether IIHWOP is nosologically distinct from IIH. The incidence, prevalence, and the degree of association with the world-wide obesity epidemic is unknown. Establishing a diagnosis of IIHWOP can be challenging, as often lumbar puncture is not routinely part of the work-up for refractory headaches. There are published diagnostic criteria for IIHWOP; however, some report uncertainty regarding a pathologically acceptable cut off for a raised lumbar puncture opening pressure, which is a key criterion. The literature provides little information to help guide clinicians in managing patients with IIHWOP. Further research is therefore needed to better understand the mechanisms that drive the development of chronic daily headaches and a relationship to intracranial pressure; and indeed, whether such patients would benefit from therapies to lower intracranial pressure. The aim of this narrative review was to perform a detailed search of the scientific literature and provide a summary of historic and current opinion regarding IIHWOP.

## 1. Introduction

The neurological disability of headache ranks second to only to stroke [1]. The pseudotumor cerebri syndrome (PTCS) embraces disorders characterised by raised intracranial pressure (ICP) [2], where the commonest symptom is headache (90%) [3]. The most prevalent of these disorders is idiopathic intracranial hypertension (IIH), which is typically found in women of child-bearing age, who are obese and where papilloedema may be sight-threatening [4,5,6]. The obesity epidemic has contributed to the increased incidence of IIH, which is approximately 15 per 100,000 in the UK [6,7,8]. IIH without papilloedema (IIHWOP) is a less well recognised condition, but is specifically recognised as a category of headache by the International Headache Society classification in 2004 [9], and has published diagnostic criteria [2].

Many aspects of IIHWOP remain under-researched, and in a priority setting partnership for IIH, an awareness of IIHWOP was within the scope of both professionals and patients [10]. Although the majority of patients with IIHWOP are described as phenotypically similar to those with IIH, it remains uncertain as to whether IIHWOP is nosologically distinct from IIH. It could be postulated that chronic daily headache (CDH), which occurs in approximately 4% of the population and 6.8% of individuals with morbid obesity could be undiagnosed IIHWOP. There are currently no recommended treatment options for IIHWOP.

In this narrative review, we have summarised the literature on headache in the presence of elevated cerebrospinal fluid (CSF) opening pressure, in the absence of papilloedema, to include the initial description, epidemiology, proposed aetiology, common clinical features, diagnosis, and potential future management options.

## 2. Materials and Methods

We searched the following databases: Embase <1980 to 2020 Week 20> and Ovid MEDLINE(R) <1946 to May Week 2 2020. The following were search terms for articles (Appendix A) in all year ranges with multiple combinations of search terms, including “Idiopathic intracranial hypertension without papilloedema”, “Pseudotumor cerebri without papilloedema”, “Pseudotumor cerebri”, “headache”, “chronic headache”, “refractory headache”, “migraine”, “intracranial hypertension”, “pulsatile tinnitus”, “raised intracranial pressure”. A further search was performed between May week 2 2020 and March week 2 2021 to ensure no further articles were required to be included. Language was restricted to English only and articles chosen were based on relevance to topics covered in this review.

## 3. Historical Evolution of Idiopathic Intracranial Hypertension without Papilloedema

In 1972 Lipton and Michelson described a 24-year-old obese lady, who presented with a 6-month history of severe daily headache with diplopia [11]. There was a partial 6th cranial nerve palsy. There was no evidence of papilloedema. Blood tests, skull x-ray, and pneumoencephalogram were normal. The lumbar puncture (LP) opening pressure (OP) was >55 cm CSF, with normal CSF constituents. Following LP, her headache improved and by one month there was no diplopia reported. At one and a half years she was completely well. The authors concluded that the clinical presentation, evidence of raised ICP and normal brain imaging best fit with a diagnosis now named IIHWOP. 

In 1991 Marcelis and Silberstein then described the first case series of IIHWOP with 8 females and 2 males who had a mean age of 37.5 years (range 26–46 years). Of these, 8 out of 10 (80%) were noted to be obese (>25% ideal body weight). Three had a history of head trauma, and one of meningitis. The LP OP ranged from 23 to 45 cm and one, non-obese patient had a LP OP > 25 cm. All patients had CDH with migrainous features and complained of headache on waking. Four out of ten (40%) reported pulsatile tinnitus. On eye examination, none of the patients had papilloedema, but 9 out of 10 (90%) had spontaneous venous pulsations. Brain imaging identified an empty sella in 2 out of 10 (20%) of cases. Three out of ten (30%) had CSF divergent surgery and the remainder were managed medically. Patients were followed up between 2 and 10 years and none consequently developed papilloedema [12].

Subsequently, other authors have reported case series of headache and elevated LP OP without the presence of papilloedema, noting between 10–15% of those with chronic migraine-like headaches have elevated LP OP (above 25 cm CSF) without papilledema [13,14]. Mathew et al. reported 85 patients with refractory transformed migraine type of CDH of which 14% had raised LP OP ranging between 23 and 45 cm CSF. In the 12 cases of elevated LP OP, the majority of these were women (10/12), with half of the women and both men (7/12) being noted to be obese. In these cases, 4 out of 12 experienced a transient improvement in headache post-LP. At follow-up none of the patients developed papilloedema [13].

Wang et al. [15] compared the clinical features of 25 patients with CDH with elevated LP OP without the presence of papilloedema with 60 control patients with CDH who had normal LP OP. Those with elevated LP OP without the presence of papilloedema had a greater body weight than controls and displayed higher LP OP (range 20 to 55 cm CSF). Univariate predictors of elevated opening pressure without the presence of papilloedema included: pulsatile tinnitus, tinnitus, blurred vision, obesity, and seizures. Multivariate analysis by stepwise logistic regression found that pulsatile tinnitus and obesity were the most important predictors in these patients diagnosed with CDH and elevated LP OP without the presence of papilloedema [15].

While these series have provided a framework for understanding the entity IIHWOP, it could be considered that not all the cases would fulfil the contemporary understanding of IIHWOP. Following these case series and case reports, the International Headache Society designated headache attributed to elevated ICP, in the absence of papilloedema, as IIHWOP [9,16]. In 2013 Friedman et al. published diagnostic criteria for PTCS with and without papilloedema in adults [2], which will be discussed below.

## 4. Measuring Intracranial Pressure

The most common and traditional method of measurement of ICP is a LP performed in the left lateral decubitus position [17]. LP is a relatively invasive procedure where up to one third can report a post-procedural complaint [18], and patients often recall a negative experience of having an LP [19]. The reported medical complication rate is low [18]. The LP OP is known to be independently influenced by the patient’s age, gender, and body mass index [20]. The level of the OP that is regarded as a normal or pathological is debated amongst clinicians [2]. 

Direct measurement of ICP is the gold standard for accuracy but is invasive, requires neurosurgical expertise, and has an inherent risk of complications [21]. Advances in neurosurgical hardware now allow for accurate “beat to beat” measurement of ICP through telemetric monitoring [21], and are recently providing unique insights into IIH [22,23]. Progress is being made in non-invasive ways to act as a surrogate measure of ICP in those with IIH including optical coherence tomography [24,25,26] and ultrasound, as are further discussed in Section 7.

## 5. Normal Lumbar Puncture Opening Pressure

The upper limit of normal LP OP is uncertain. Whiteley et al. [27] showed that a 95% reference interval for LP OP was 10 to 25 cm CSF. The International Headache Society defined the upper limit of normal LP OP to be 20 cm in their recommendations (2004) [16]. This was further refined in 2018 to be 25 cm for non-obese children, and 28 for obese children [9]. A recent large population-based study showed that LP OP can vary significantly among individuals. Higher OPs were associated with higher BMI and younger age. The study reported that 15.8% of normal individuals had an OP ≥ 200 mmH_2_0 and 2% had an OP ≥ 200 mmH_2_0. In addition, they reported a group of 79 patients who had two LPs within 2.5 years and in 12.7% there was a difference of ≥50 mmH_2_O [15].

Another consideration is the normal range of LP OP in those previously diagnosed with raised ICP. In 1983 Corbett et al. [28] measured LP OP in patients with chronic IIH and noted that 28% (5/18) of their series had LP OP > 25 cm CSF, despite not having papilloedema. One patient had a LP OP of 34 cm CSF, which when reassessed one month later had fallen to 14 cm CSF. A single measure LP is a “snap-shot” of the ICP in the lumbar compartment, and if mildly elevated could reflect altered brain compliance.

## 6. Diagnostic Criteria for IIHWOP

The International Headache Society currently do not detail diagnostic criteria for IIHWOP. They define headache attributed to IIH as when a person with new onset headache or significant worsening of headache has been diagnosed with IIH; or the CSF pressure exceeds 250 mm CSF (or 280 mm CSF in obese children) and the headache is accompanied with either or both pulsatile tinnitus and papilloedema [16].

In 2013 Friedman et al. [2] recommended that a diagnosis of PTCS without papilloedema could be made in patients who have normal brain imaging including magnetic resonance venography (MRV) or computed tomography venography (CTV) (apart from radiological signs of raised ICP); normal neurological examination (except cranial nerve abnormalities); normal CSF constituents; LP OP > 25 cm CSF; and a unilateral or bilateral 6th nerve palsy. In the absence of 6th nerve palsy, a diagnosis of PTCS without papilloedema could still be suggested in patients who fulfil the criteria for IIHWOP, and in the presence of three out of four neuro-radiological signs of raised ICP. The neuroimaging features included: empty sella; flattening of the posterior aspect of the globe; distension of the perioptic subarachnoid space with or without a tortuous optic nerve; and transverse venous sinus stenosis. When road-tested, these MRI imaging criteria were to help distinguish those patients with PTCS without papilledema, from those with chronic daily headache (CDH) or migraine headache and an incidentally elevated LP OP [29].

The diagnostic criterion of LP OP > 25 cm CSF was debated by De Simone et al. [30], who suggested that cases of IIHWOP could be missed, as this level was too high. They suggested that presence of bilateral transverse venous sinus abnormalities in headache patients in the absence of papilloedema was predictive of raised ICP, as was a symptomatic improvement in headache following LP as evidenced by other clinician’s opinions in the literature [12,13,30,31,32,33,34]. They recommended that the cut off for the LP OP for a diagnosis of IIHWOP should be 20 cm CSF. Recently IIH patients have reported a mixed response of either exacerbation or relief of headache following LP [35]. Indeed, evidence has shown that headache relief following an LP is not only experienced by those with raised ICP, but also in a quarter of migraine controls in one study [36]. This evidence would challenge the concept of patient reported benefit following an LP, as a diagnostic tool for IIHWOP.

In 2018, a multi-disciplinary group of ophthalmologists, neurologists, and neurosurgeons in the United Kingdom agreed that in the context of IIH there was not one single cut off point at which the condition should be diagnosed and that the zone between the measures of 25 and 30 cm CSF could be normal and not causative of IIH for an individual [4]. It was considered that increasing levels of the LP OP would be thought to be pathological [5]. These guidelines recommended that LP OP should not be taken in isolation and should be considered within the clinical context [4,5].

## 7. Does Intracranial Pressure Monitoring Have a Role in the Diagnosis of IIHWOP?

The diagnosis of IIHWOP currently depends on a single raised LP OP measurement. Two studies have shown that a single measurement of ICP by LP may miss patients with potential IIHWOP [37,38]. Torbey et al. reported on the use of 24 h continuous pressure monitoring via a lumbar drain in 10 patients with IIHWOP [37]. The majority of patients were female and obese. None of patients reported pulsatile tinnitus. Spontaneous venous pulsations were identified in 50%. LP OP prior to continuous monitoring ranged from 21.0 to 39.8 cm CSF. Forty percent (4/10) had LP OP < 25 cm CSF. During continuous monitoring all patients displayed both normal and abnormal pressure waves. All patients displayed abnormal pressure during sleep which included B waves, plateaus or near plateaus. Eighty percent (8/10) of patients benefited from CSF removal. All patients received CSF divergent surgery. In 2010, Bono et al. identified abnormal CSF pressure waves in headache patients with MRV evidence of bilateral transverse sinus stenosis (TSS), but not in patients with unilateral TSS or normal MRV, who had CSF pressure monitored for 1 h [38]. Thirty-five percent (18/38) of patients had raised LP OP and abnormal pressure waves. Eighty-seven percent (26/48) of the patients with normal LP OP had abnormal pressure waves. The health controls and headache controls with unilateral TSS did not display abnormal pressure waves during monitoring [38]. These studies indicate that a single LP OP measurement may not be reflective of the average ICP, and do not provide the detail of the waveform patterns observed of ICP.

Telemetric ICP monitors have the ability to longitudinally record ICP. Although the current limitation is that they are invasive to site, they are becoming more frequently used. They provide measurements of ICP with the patient in their normal environment undertaking normal activities of daily living, and may in the future provide greater insights in both the healthy and those with altered ICP conditions [21].

## 8. Non-Invasive Indicators of Raised ICP

Indirect methods of detection of raised ICP include non-specific neuro-imaging features, more experimental methods such as magnetic resonance imaging (MRI)-ICP, and ocular optical coherence tomography [21]. A number of case series have detailed the neuroimaging features in conjunction with the clinical characteristics in IIHWOP, mainly focusing on TSS.

### 8.1. Neuro-Imaging Features Associated with Raised ICP

The neuroimaging features associated with raised ICP included: partial empty sella; empty sella; flattening of the posterior aspect of the globe; distension of the perioptic subarachnoid space with or without a tortuous optic nerve; and transverse venous sinus stenosis. These signs may be seen in normal individuals and are non-specific. The 2013 revised criteria for PTSC recommend 3 or more neuro-imaging criteria being present in order to make a probable diagnosis of IIH without papilloedema, where there is no sixth nerve palsy and when all the other criteria for PTSC are met [2]. This has been recently validated. A combination of any 3 of 4 MRI features was found to be nearly 100% specific, with a sensitivity of 64%, for a diagnosis of IIH without papilloedema in patients with chronic headache, no papilloedema, and elevated lumbar puncture opening pressure. In isolation no individual MRI feature of intracranial hypertension had sufficient specificity to be diagnostic of raised intracranial pressure. Reduced pituitary gland height (less than 4.8 mm) was moderately sensitive at 80% but had a low specificity of 64% [31]. In addition, no single MRI characteristic associated with PTSC has been found to be predictive of visual outcomes [39].

### 8.2. Transverse Sinus Stenosis, Chronic Daily Headache and IIHWOP

Venography (MRV or CTV) has shown that raised ICP in unilateral or bilateral TSS is common in those with IIH [40]. Asymptomatic bilateral TSS exists in patients with LP OP ≤25 cm CSF, but was uncommon [41]. Bilateral TSS is present in the majority of patients IIHWOP [30,31,32,33,42].

Bono et al. demonstrated that the presence of bilateral TSS in patients with migraine was predictive of IIHWOP [31]. Seven hundred and twenty-four patients with a diagnosis of migraine underwent MRV. Of these, 675 (93%) had normal MRV. Of these, 70 patients had normal LP OP. Forty-nine out of 724 (6.9%) patients demonstrated bilateral TSS on MRV. Of these, 28 had a LP and 19 (18 females, 1 male) out of 28 (67.8%) had LP OP > 20 cm and were diagnosed with IIHWOP. The headache profiles of patients with IIHWOP did not differ from migraine patients who did not have raised LP OP. Patients with MRV evidence of bilateral TSS and LP OP > 25 cm had significantly raised body mass index compared to migraine patients with normal MRV. Headaches improved in 13 out of 19 (68.4%) IIHWOP patients following a LP. In chronic tension-type headache patients, bilateral TSS was found in 18 out of 198 (9%) patients [32]. Of these, 13 had an LP and 9 of those 13 (69.2%) had LP OP > 20 cm CSF, and the authors suggested these patients should have been diagnosed with IIHWOP. LP OP was normal in all who had a normal MRV. Patients with bilateral TSS and raised LP OP were all female and were significantly more obese than patients with normal MRV. Headache improved in 8 out of 9 (89%) of IIHWOP patients who had a LP [31]. In 2010 Bono et al. looked at the upper limit of normal LP OP in 217 headache sufferers in relation to presence of bilateral TSS [43]. Of the 217 patients, 50 (23%) patients had MRV evidence of bilateral TSS, 47 out of 217 (17%) unilateral TS and 65 out of 217 (55%) no TS. Twenty-four out of 217 (11%) patients had LP OP > 20 cm CSF. Of these, all 24 patients had evidence of bilateral TSS. Twenty-six of 193 (13%) patients with LP OP < 20 cm had evidence of bilateral TSS. Of these, 2 patients on repeat LP had LP OP > 20 cm [43]. As with previous studies, patients with bilateral TSS were predominantly obese females.

In 2018 Bono et al. examined CSF pressure-related features in 148 patients with CDH [33] (Table 1). All patients underwent one hour CSF pressure monitoring by LP and MRV. High pressure or abnormal pressure waves were observed in 93 out of 148 (63%) of patients. Of these, 57 of 93(60%) had LP OP between 20 and 25 cm, while 37 out of 93 (40%) of patients had LP OP > 25 cm. Patients with LP OP > 25 cm had more severe abnormal pressure pulsations and more severe headache, which were more often nocturnal and associated with posture [33].

Bono et al. [44] demonstrated that bilateral TSS could persist in IIH after normalization of the ICP. Fourteen patients with IIH and evidence of bilateral TSS were followed over a 6-year period. Patients had repeated MRV followed by LP. Bilateral TSS persisted in all patients. In 9 out of 14 (64%) of patients the LP OP normalized [44].

Quattrone et al. [42] investigated the incidence of cerebral venous thrombosis (CVT) and isolated IIHWOP in 114 patients with CDH. Of these 114 patients, 11 (9.6%) patients with CDH had evidence of unilateral (6 out of 11 (55%), 5 females, 1 male) or bilateral CVT (5 of 11 (45%), 5 females) on MRV imaging. This proportion of patients with CVT was considered high, and may have been due to the use of MR venography and the interpretation of flow in the transverse sinus, coupled with a hypoplastic sinus. LP OP was measured in 27 out of 103 patients with normal MRV and patients with uni- or bilateral CVT. LP OP was significantly higher in patients with bilateral CVT compared to unilateral CVT or CDH patients with normal MRV. Patients with bilateral CVT were all female and displayed higher BMI than patients with unilateral CVT or controls, but this did not reach significance. Their headache profiles did not differ between groups [40].

### 8.3. Detection of Raised ICP by Spontaneous Venous Pulsations

Spontaneous retinal venous pulsation (SVP) is a subtle variation in the calibre of the retinal vein(s) as they cross the optic disc [45]. Ten percent of the normal population do not have detectable SVP [46,47]. SVP can be detected clinically on examination, and has been recorded using video ophthalmoscopy and optical coherence tomography (OCT) imaging. The relationship between ICP and SVP is of interest in developing a non-invasive measure of ICP at the optic nerve head. The tree shrew is proving to be a promising model [48]. Using OCT, the absence of SVP on retinal infrared video recordings is associated with higher levels of ICP and should raise the suspicion of intracranial hypertension in those with papilloedema [49]. Critics may consider OCT, and ophthalmic clinical examination to be redundant in IIHWOP; however, Halsey presented a case with no papilledema and no SVP, whom he suspected had IIHWOP and following LP, SVP was evident on clinical examination [50].

### 8.4. Detection of Raised ICP by Optic Nerve Sheath Diameter (ONSD) Changes

A number of studies have reported that measuring the optic nerve sheath diameter (ONSD) using imaging technologies (such as CT and MRI) provides an indicator for raised ICP [51,52]. A recent meta-analysis found that optic nerve sheath diameter as measured by ultrasound had high diagnostic accuracy (estimated sensitivity of 90%; and specificity of 88%) as compared to CT ONSD (estimated sensitivity of 93%; and specificity of 79%) and MRI ONSD (estimated sensitivity of 77% and specificity of 89%) [53]. Ultrasound ONSD has yet to be explored in IIHWOP.

## 9. Incidence of Documented Raised ICP in the Presence of Headache

Several studies have examined the incidence of raised ICP in different cohorts of headache patients [13,14,30,31,32,33,54] (Table 2). Of particular note is the large normal population study by Wang et al. [29] investigating LP OP in volunteers. While some within their cohort had elevated LP OP and headache, the portion of those with a higher LP OP had less headache. Whilst the headache was not classified within this study, the principle of understanding the difficulty in using LP OP as a sole criterion for diagnosing IIHWOP is raised.

Mathew et al. identified 14% of patients with refractory transformed migraine type of CDH with LP OP > 20 cm [13]. In separate studies, Bono et al. [31,32,33,38] examined patients with migraine, tension-type headache, “headache sufferers”, and CDH, finding LP OP > 20 cm 4.6% in migraine and 6.2% in tension-type headache. For “headache sufferers”, 11% had LP OP > 20 cm [40] and 63% for CDH patients with LP OP > 20 cm [33]. Twenty-five percent of CDH patients had LP OP > 25 cm [33]. Vieira et al. [14] identified 10% of chronic migraine patients with LP OP > 20 cm, while De Simone et al. [30] identified 86%. In De Simone et al. [30] case series at 4-month follow-up, the chronic pain had reverted to episodic migraine in 17 out of 44 (38.6%). In the most recent study, Favoni et al. [34] identified 22% of patients with CDH with LP OP > 20 cm and 5% > 25 cm. However, De Simone suggested that his cohort differed from Favoni’s in that they were less likely to have confounding medication-overuse type headache [54].

## 10. Incidence of IIHWOP, in Accordance with the Diagnostic Criteria

In institutional based series of those with chronic refractory headache, the frequency found of IIHWOP has been reported to range between 2.5% to 86% [14,29,33]. This wide range appeared to depend on a number of factors. Firstly, the majority of the publications are prior to the publication of the diagnostic criteria for IIHWOP in 2013 (Friedman 2013), when IIHWOP was generally used to describe those with headache and a raised LP OP > 20 cm CSF, rather than LP OP > 25 cm CSF. The others include the population in which they are determining their cohorts, from headache clinics, or those with refractory daily headache, and the proportion of those with medication overuse headaches.

Studies that post-date the criteria include Favoni et al. [34] who examined the incidence of IIHWOP in 40 patients with CDH: 9 out of 40 (22%) had LP OP > 20 cm, while 2 out of 40 (5%) had LP OP > 25 cm. Patients with LP OP > 20 cm were more likely to be obese females. After applying Friedman’s criteria, only 1 out of 40 (2.5%) of patients could be classified as IIHWOP. These authors have debated the level of LP OP at >25 cm CSF as too stringent [54].

## 11. How Often Is Papilloedema *Completely Absent* in Chronically High ICP?

There are few studies to help answer this question. One study found in evaluating patients with IIH that about 6% of patients with chronically high ICP have no papilloedema (IIHWOP) [13]. In this study, IIHWOP patients reported symptoms of photopsias and unexpectedly had non-physiological visual field constriction as compared to those with typical IIH [55]. Future studies using OCT imaging of the optic nerve head could help evaluate this question.

## 12. Why Is Papilloedema Not a Universal Clinical Feature of Raised ICP?

The optic nerve sheath subarachnoid spaces (ONS-SAS) are often seen to be distended on neuroimaging studies in patients with IIH. The ONS-SAS have been considered to be in free communication with the suprasellar cistern and intracranial subarachnoid spaces (IC-SAS), which would suggest that as ICP rises, papilloedema should develop. However, anatomically the ONS-SAS are extremely narrow and have numerous trabeculae and septae that reduce both CSF volume and CSF flow. Killer et al. [56] performed detailed examinations on six patients with optic nerve head swelling (three with asymmetric papilloedema) and hypothesized that if the ONS-SAS were freely communicating with the IC-SAS, then the distribution of CSF proteins would be similar. They disproved the hypothesis by finding that CSF obtained from the ONS-SAS following optic nerve sheath fenestration demonstrated markedly different biomarker concentrations compared with those in the CSF obtained from the corresponding lumbar CSF [56].

The threshold of CSF pressure required to develop papilloedema may depend on individual patient characteristics and it is possible that those with IIHWOP have a threshold of ICP that is higher than others [12]. Indeed, among adult patients with LP OP above the accepted upper limits of normal, those with papilloedema have been found to have higher LP OP than those without papilloedema [51]. This finding was also reported in a paediatric population [57]. In addition, Faz et al. (2010) [58] reported that only half of the children (13/27) had papilloedema at the initial baseline examination, where the mean LP OP was 310 mm CSF (range 210 mm CSF to 650 mm CSF with a mean of 319 mm CSF). Those without papilloedema had a mean LP OP of 317 mm CSF (range 230 mm CSF to 470 mm CSF). However, it was suggested that the papilloedema may have not have yet developed at the time of the examination, as no follow-up examination was detailed in Faz et al. [58]. Detailed longitudinal monitoring for papilloedema, in the presence of documented raised ICP, is a limitation of reported case series [59].

In an animal study, where an inflated balloon in the subarachnoid space was used to induce raised ICP, papilloedema developed in 30% within 24 h and in 90% within 5 days. The faster the balloon was inflated and the higher the ICP, the greater the likelihood and degree of papilloedema [60]. However, in acute traumatic intracranial haemorrhage or ruptured aneurysm with continuous ICP monitoring revealing raised ICP and where fundoscopy was performed twice a day for 7 days, only one patient developed a blurred disc margin at day 6 [61]. Other studies have affirmed a low prevalence of papilloedema in both significant head trauma [62] or rupture intracranial aneurysm [63] when fundoscopy was performed within a few days of the incident.

Anatomical anomalies within the optic nerve sheath have been postulated which might prevent the development of papilloedema [12,64] Unilateral papilloedema can be present in IIH [60] and a portion of patients have a 2 grade difference in their papilloedema (as measured using the Frisen grading system), and this may be considered to be the reason why papilloedema has yet to develop in some with IIHWOP. IIH can present asymmetric papilloedema and MRI characteristics have been identified that explain the asymmetry [65]. Bidot et al. [66] proposed that a narrower optic canal is associated with less disc oedema because of restricted CSF flow from the brain to the optic nerve sheath, resulting in lower intraorbital CSF sheath pressure. However, using CT imaging, which is known to be more accurate in measuring bone architecture, there was no significant correlation between optic canal measurements and the grade of papilloedema in both IIH and papilloedema at baseline, and in a control group [67,68].

## 13. Management of IIHWOP

The management of IIHWOP remains unclear and is frequently unsatisfactory. The majority of case series indicated that headaches improved following medical treatment with acetazolamide and topiramate, and intervention by repeated CSF drainage by LP [12,31,32,33,34,54,64]. Commonly, reduction in headaches was not sustained and in some, repeated LP was complicated by CSF leaks and low pressure headaches [63]. Unlike IIH there does not appear to be a threat of visual loss in IIHWOP [64], and hence why caution in treatments that are invasive or have significant adverse profiles should be avoided. Therefore, CSF diversion surgery which is known to be surgically more challenging in IIH as compared to other shunts, may not be the optimal management option [4]. Indeed, when compared to CSF shunts for all aetiologies, shunts for IIH are associated with significantly higher revision rates and complications [69]. In addition although headache can improve in the short term, longitudinal data suggests it is not an ideal therapy for headache [70]. Despite this, CSF divergent surgery has been adopted in a number of series and resolution of headache has been reported [12,37,71].

Endovascular stenting of TSS has also been deployed in management of IIHWOP [72,73]. However, TSS has also been shown to persist in patients with IIH and IIHWOP despite normalization of ICP [44], indicating that other factors are likely to be involved in mechanisms which raise ICP.

Medical management appears to be largely unexplored. As the phenotype of raised ICP headache is for some not classically an early morning headache [74], but can mimic migraine, future studies may investigate the utility of mainstream headache medications for management [4,75]. Indeed, discoveries of therapies for headache attributable to IIH also require systematic evaluation, as there is only one open label study investigating use of a calcitonin gene related peptide monoclonal antibody to treat headache in patients with active IIH [76]. What was interesting was that a small subset of patients had reoccurrence of their papilloedema without a return of their headache symptoms, which may implicate CGRP in the underlying pathways causing headache [77]. Likewise, the role of weight management requires further exploration as obesity is a consistent risk factor in the majority of IIHWOP case series, and for IIH this is a modifiable risk factor [78].

## 14. Future Research

Effective research programs and well-designed studies in IIHWOP are required to delineate the clinical phenotype, and establish the risk factors driving the underlying pathophysiology. An understanding of the link between raised ICP and development of headache is beyond the scope of this review. Some have suggested that there is little relationship between LP OP and headache frequency and severity. The mechanisms which link raised ICP to development of headache are not fully understood. One possibility is that in genetically-predisposed individuals, with marginally raised ICP (20–30 cm CSF), the threshold to trigger headache attacks is lower and this mechanism drives daily headache. Proving this in such patients with CDH refractory to migraine treatment may guide future management strategies and support further research into the use of drugs that reduce ICP.

Research in IIHWOP should involve identification of specific CSF and serum biomarkers that could contribute to understanding the pathogenesis of IIHWOP, and provide an accurate diagnostic tool. This would permit a shift from clinical diagnostic criteria to an accurate test, permitting an accurate measurement of the incidence and prevalence of the disease. In a meta-analysis of biochemical measurements in CSF and blood from chronic and episodic migraine patients, a total of 62 unique compounds were reported to have been measured in CSF from migraine patients. For example, glutamate, calcitonin gene-related peptide, and nerve growth factor concentrations were found to be increased, and b-endorphin concentrations decreased. These changes were also reported to be present in blood, with the exception of nerve growth factor. [79] Biochemical signatures of disease of the central nervous system, in both the CSF and blood, have the potential to identify clear biomarker targets for future diagnostic studies for different classifications of headaches. This would be important in the distinction between primary headache disorders and those secondary headache disorders caused by raised ICP.

The current treatment options in IIHWOP are limited and future studies are required to establish whether headache therapies as well as CSF-production lowering drugs may benefit these patients, addressing the most disabling symptoms of headache.

## 15. Conclusions

The considerable variation in the incidence and prevalence of IIHWOP reported in the literature probably reflects the uncertainty of what level of LP OP is regarded as diagnostic. Some suggest that the present diagnostic criteria for IIHWOP are not sensitive enough to capture all patients, due to the cut off of LP OP at >25 cm CSF. The clinical relevance of lower levels of ICP in patients with headache remains less clear. The pathogenesis and nosological position of IIHWOP in relation to IIH remains relatively unexplored and requires further detailed evaluation. Patients with both conditions are typically young obese females and it is therefore tempting to conclude that anatomical differences relating to the optic nerve and therefore the threshold at which papilloedema develops, separate the two. The presence of TSS in a high proportion of patients with IIHWOP is noteworthy and when identified in patients with chronic headache may support further assessment of ICP. Treatment options for headache in IIHWOP require development. Further research is therefore needed to better understand the mechanisms that drive the relationship between mildly raised ICP and development of headache and whether such patients would benefit from therapy to lower ICP.

## Figures and Tables

**Table 1 life-11-00472-t001:** Summary for MRI/MRV features reported in Bono et al. [33].

Lumbar Puncture Opening Pressure	Neuroimaging Feature	%	Number
>25 cm CSF	Bilateral Transverse Sinus Stenosis	78%	29/37
	Empty sella	68%	25/37
	Perioptic subarachnoid space distension	57%	21/37
20–25 cm CSF	Bilateral Transverse Sinus Stenosis	71%	40/56
	Empty sella	57%	32/56
	Perioptic subarachnoid space distension	27%	15/56
<20 cm CSF	Bilateral Transverse Sinus Stenosis	2%	1/55
	Empty sella	7%	4/55
	Perioptic subarachnoid space distension	2%	1/55

**Table 2 life-11-00472-t002:** Studies detailing characteristics of IIHWOP in different headache populations.

Study	Headache Phenotype as Reported by Authors	Study Population Number	IIHWOP							
			Number with IIHWOP	% Lumbar Puncture Opening Pressure > 20 cm CSF	% Lumbar Puncture Opening Pressure > 25 cm CSF	Female:Male	Age in Years (Range or Standard Deviation [±] Where Reported)	Obesity Description or Body Mass index (kg/m^2^) (Range or Standard Deviation [±] Where Reported)	Mean Lumbar Puncture Pressure (Range or Standard Deviation [±] Where Reported)	% Improved Post-LP
1996 Mathew et al. [13]	refractory transformed migraine type of chronic daily headache	85	12	14.1%	12.9%	10:2	34 (13–54)	100% males obese, 50% females obese	31.2 (23.0–45.0) cm CSF	33%
2001 Quattrone et al. [40]	chronic daily headache	114	5	4.4%	1.8%	5:0	31.6 (18–46)	33.1 (28.7–43.4) 80% obese	26.4 (22.4–31.5) cm CSF	..
2006 Bono et al. [31]	chronic migraine	724 (98 had LP)	19	* 4.5%	* 4.1%	18:1	35.0 ± 7.2	31.7 ± 3.9	282.5 ± 40.6 mm CSF	68.4%
2008 Vieira et al. [14]	chronic migraine	60	6	10%	5%	6:0	41.2 (26–52)	29.5 (21.8–33.3) 50% obese	26.2 (24.4–30) cm CSF	100% immediate
2008 Bono et al. [32]	chronic tension-type headache	198 (58 had LP)	9	* 6.3%	* 4.2%	9:0	40.8 ± 6.6	33.7 ± 7.6	270. 4 ± 34.1 mm CSF	89%
2010 Bono et al. [38]	Headache sufferers	98	18	18.3%	..	17:1	39.6 ± 13.6	29.9 (21.4–40.8)	24.7(20.5–32.9) cm CSF	“majority” of patients had benefit reported at 2–4 weeks
2014 De Simone et al. [30]	Unresponsive chronic migraine	44	38	86%	43%	Study population mainly female	Study population 37.5 (33–40)	Study population 56.8% overweight/obese		38.7% at 4 months
2018 Bono et al. [33]	chronic daily headache	148	93	63%	25%	80:13	$40.1 ± 15.1 $	$32.5 ± 5.7	$25.6 ± 20.1 cm CSF	100%
2018 Favoni et al. [34]	chronic daily headache	40	9	22%	5%	8:1	50 ± 8	32 (25–38)	24.5 (21.1–25.8)	78% at 1 month

* estimate based on multiple calculations; $ pooled mean ± SD.

## Data Availability

Search strategies are within the Appendix A. However further reasonable requests for the data will be met by the corresponding author, and if agreed a data sharing agreement will be required.

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
