# Peer review of "Current Perspectives on Idiopathic Intracranial Hypertension without Papilloedema"

_life, 2021, doi:10.3390/life11060472_

Round 1

Reviewer 1 Report

The authors provide an excellent review of IIH without papilledema (IIHWOP).  This is well structured and informative.  Below are some comments:

Much of the manuscript suggests that IIHWOP is being underdiagnosed. Please discuss the caution of overdiagnosis as well, for example the study out of Emory that found that 39.5% referrals referred for IIH did not have the disease. This can lead to treatments that can have significant side effects, which is especially pertinent because there is no threat to the vision in IIHWOP.  Also emphasize that a definitive diagnosis of IIHWOP is hard to make.  Friedeman 2013 diagnostic criteria states that a diagnosis of IIH can only be suggested without the presence of papilledema or 6th nerve palsies.

Abstract:

Consider ending the abstract with a sentence saying what this manuscript is doing, which is a review of IIHWOP.

Introduction:

Intro discusses possible mechanisms of headache, including triggering migraine. This is not discussed elsewhere and does not appear to be the main point of the review article so can consider modifying.

3. Historical evolution of IIHWOP:

In reviewing the literature, please consider adding commentary as to whether the patients in the case series truly had IIHWOP and caveats to their data. For example, Mercelis and Silberstein, 1991’s original series, one could argue that many of these patients did not have IIHWOP since only 20% had empty sella and only 40% had pulse synchronous tinnitus. Caution should be made to made a diagnosis of IIH based on opening pressure alone.

“Subsequently, other authors have reported case series of headache and elevated LP OP without the presence of papilloedema, noting between 10-15% of those with chronic migraine-like headache have elevated LP OP without papilledema”. Is this 20cm H20 or 25cm H20 cutoff? It is important to note that many normal patients have an opening pressure >20cm H20 and some even >25. A recent study looking at the Mayo Clinic Study of Aging showed that 15.8% of normal individuals had an opening pressure >20 and 2% had an opening pressure >25. In addition, serial LP’s in these normal individuals shows that 12.7% had a difference of 5cm from year to year. Wang, Feng, et al. "Population-based evaluation of lumbar puncture opening pressures." Frontiers in neurology 10 (2019): 899.

Please provide the details of the International Headache Society definition of IIH-WOP.

4. Measuring ICP: In the section on measuring ICP, include direct ICP monitoring, which likely provides the most accurate ICP monitoring, but is invasive and requires admission.

6. Diagnostic criteria for IIHWOP:

For the Friedeman et al 2013 diagnostic criteria, a diagnosis of IIHWOP can only be “suggested” without papilledema or 6th nerve palsy, and this suggestion requires the 3 indirect signs of raised ICP. Please include “suggested” and that it is hard to make a definitive diagnosis of IIHWOP.

8.1 Neuro-imaging features associated with raised ICP:

The authors discuss venous sinus stenosis and spontaneous venous pulsations. Please also review empty sella and dilation of optic nerve sheath. Also discuss that these signs can be seen in normal individuals.

p.6: for the discussion on Quattrone et al, the original manuscript does use the term cerebral venous thrombosis, but is this truly CVT or was this actually transverse sinus stenosis? 9.6% true CVT seems high for patients with chronic daily headache. Please comment.

9. Incidence of documented raised ICP in the presence of headache:

When discussing these studies with “elevated” ICP in patients with headaches, please also compare to studies of OP in normal individuals, such as Wang et al. 2019.

  1. How often is papilledema completely absent in chronically high ICP?:

Discuss whether OCT can pick up subtle disc edema in patients felt to not have papilledema.

  1. Why is papilledema not a university clinical feature of raised ICP?:

Discuss that unilateral papilledema can be present and that 10% can have >=2 grade papilledema difference so IIHWOP can exist.

  1. Management of IIHWOP

Caution that medications and especially surgical interventions are not necessarily benign so caution must be taken to not be overly aggressive for IIHWOP since there is no threat to the vision.

Author Response

Reviewer 1

We would like to thank the reviewer for their time and helpful comments.

The authors provide an excellent review of IIH without papilledema (IIHWOP).  This is well structured and informative.  Below are some comments:

Much of the manuscript suggests that IIHWOP is being underdiagnosed. Please discuss the caution of overdiagnosis as well, for example the study out of Emory that found that 39.5% referrals referred for IIH did not have the disease. This can lead to treatments that can have significant side effects, which is especially pertinent because there is no threat to the vision in IIHWOP.  Also emphasize that a definitive diagnosis of IIHWOP is hard to make.  Friedeman 2013 diagnostic criteria states that a diagnosis of IIH can only be suggested without the presence of papilledema or 6th nerve palsies.

1 emphasize that a definitive diagnosis of IIHWOP is hard to make

  • In the abstract: Establishing a diagnosis of IIHWOP can be challenging, as often lumbar puncture is not routinely part of the work-up for refractory headaches

2 Consider ending the abstract with a sentence saying what this manuscript is doing, which is a review of IIHWOP.

Added:

  • The aim of this narrative review was to perform a detailed search of the scientific literature and provide a summary of the historic and current opinion regarding IIHWOP.

  1. Introduction: Intro discusses possible mechanisms of headache, including triggering migraine. This is not discussed elsewhere and does not appear to be the main point of the review article so can consider modifying.
  • Removed thanks to the future research section.

  1. Historical evolution of IIHWOP:

In reviewing the literature, please consider adding commentary as to whether the patients in the case series truly had IIHWOP and caveats to their data. For example, Mercelis and Silberstein, 1991’s original series, one could argue that many of these patients did not have IIHWOP since only 20% had empty sella and only 40% had pulse synchronous tinnitus. Caution should be made to made a diagnosis of IIH based on opening pressure alone.

We agree with your comments and have added

  • Last paragraph to section 3. While these series have provided a framework for understanding the entity IIHWOP, it could be considered that not all the cases would fulfil the contemporary understanding of IIHWOP.
  1. Subsequently, other authors have reported case series of headache and elevated LP OP without the presence of papilloedema, noting between 10-15% of those with chronic migraine-like headache have elevated LP OP without papilledema”. Is this 20cm H20 or 25cm H20 cutoff?

We have added

  • The cut of was 25. We have added: Subsequently, other authors have reported case series of headache and elevated LP OP without the presence of papilloedema, noting between 10-15% of those with chronic migraine-like headaches have elevated LPOP (above 25 cm CSF).
  1. It is important to note that many normal patients have an opening pressure >20cm H20 and some even >25. A recent study looking at the Mayo Clinic Study of Aging showed that 15.8% of normal individuals had an opening pressure >20 and 2% had an opening pressure >25.

In addition, serial LP’s in these normal individuals shows that 12.7% had a difference of 5cm from year to year. Wang, Feng, et al. "Population-based evaluation of lumbar puncture opening pressures." Frontiers in neurology 10 (2019): 899.

We agree, we have added:

A recent large population-based study showed that LP OP can vary significantly among individuals. Higher OPs were associated with higher BMI and younger age. It reported that 15.8% of normal individuals had an OP ≥ 200 mmH20 and 2% had an OP ≥ 200 mmH20. In addition, they reported a group of 79 patients who had two LPs within 2.5 years and in 12.7% there was a difference of ≥50 mmH2O [22].

  1. Please provide the details of the International Headache Society definition of IIH-WOP. They don’t have a special section on IIHWOP.They have headache attributed to IIH, where you either need pulsatile tinnitus or papilledema.

To clarify this, we have written:

  • Diagnostic criteria for IIHWOP
  • The International Headache Society currently do not detail diagnostic criteria for IIHWOP. They define headache attributed to IIH where a person with new onset headache or significant worsening of headache has been diagnosed with IIH; or the CSF pressure exceeds 250mmCSF (or 280mmCSF in obese children) and the headache is accompanied with either or both pulsatile tinnitus and papilloedema [16].

  1. 4. Measuring ICP: In the section on measuring ICP, include direct ICP monitoring, which likely provides the most accurate ICP monitoring, but is invasive and requires admission.

We have added:

  • Direct measurement of ICP is the gold standard for accuracy but is invasive and requires neurosurgical expertise, and has an inherent risk of complications [22].

  1. 6. Diagnostic criteria for IIHWOP:

For the Friedeman et al 2013 diagnostic criteria, a diagnosis of IIHWOP can only be “suggested” without papilledema or 6th nerve palsy, and this suggestion requires the 3 indirect signs of raised ICP. Please include “suggested” and that it is hard to make a definitive diagnosis of IIHWOP.

We have added

  • Suggested: In the absence of 6th nerve palsy, a diagnosis of PTCS without papilloedema could still be suggested in patients who fulfil the criteria for IIHWOP, and in the presence of three out of four neuro-radiological signs of raised ICP

  1. 8.1 Neuro-imaging features associated with raised ICP:

The authors discuss venous sinus stenosis and spontaneous venous pulsations. Please also review empty sella and dilation of optic nerve sheath. Also discuss that these signs can be seen in normal individuals.

We have reordered this section to include you comments.

We have added as a summary at the start of this section:

  • The neuroimaging features associated with raised ICP included: partial empty sella; empty sella; flattening of the posterior aspect of the globe; distension of the perioptic subarachnoid space with or without a tortuous optic nerve; and transverse venous sinus stenosis. These signs may be seen in normal individuals and are non-specific. The 2013 revised criteria for PTSC recommend 3 or more neuro-imaging criteria being present in order to make a probable diagnosis of IIH without papilloedema, where there was no sixth nerve palsy and when all the other criteria for PTSC were met [2]. This has been recently validated. A combination of any 3 of 4 MRI features was found to be nearly 100% specific, with a sensitivity of 64%, for a diagnosis of IIH without papilloedema in patients with chronic headache, no papilloedema and elevated lumbar puncture opening pressure. In isolation no individual MRI feature of intracranial hypertension had sufficient specificity to be diagnostic of raised intracranial pressure. Reduced pituitary gland height (less than 4.8mm) was moderately sensitive at 80% but had a low specificity of 64%. [31] In addition no single MRI characteristics associated with PTSC have been found to be predictive of visual outcomes [40]

We have added a short section on ONSD/ICP:

  • 3 Detection of raised ICP by optic nerve sheath diameter (ONSD) changes
  • A number of studies have reported that measuring the optic nerve sheath diameter (ONSD) using imaging technologies (such as CT and MRI) provides an indicator for raised ICP [52][53]. A recent meta-analysis found that optic nerve sheath diameter as measured by ultrasound had high diagnostic accuracy (estimated sensitivity of 90%; and specificity of 88%) as compared to CT ONSD (estimated sensitivity of 93%; and specificity of 79%) and MRI ONSD (estimated sensitivity of 77% and specificity of 89%) [53]. Ultrasound ONSD has yet to be explored in IIHWOP.

  1. p.6: for the discussion on Quattrone et al, the original manuscript does use the term cerebral venous thrombosis, but is this truly CVT or was this actually transverse sinus stenosis? 9.6% true CVT seems high for patients with chronic daily headache. Please comment.

We agree, we have added to this section:

  • This portion of patients with CVT was considered high, and may have been due to the use of MR venography and the interpretation of flow in the transverse sinus, coupled with a hypoplastic sinus

  1. 9. Incidence of documented raised ICP in the presence of headache:

When discussing these studies with “elevated” ICP in patients with headaches, please also compare to studies of OP in normal individuals, such as Wang et al. 2019.

Added:

  • Of particular note is the large normal population study by Wang et al [29] investigating LP OP in volunteers. While some within their cohort had elevated LP OP and headache, the portion of those with a higher LPOP had less headache. Whilst the headache was not classified within this study the principle of understanding the difficultly in using LPOP as a sole criterion for diagnosing IIHWOP is raised.

  1. How often is papilledema completely absent in chronically high ICP?:

Discuss whether OCT can pick up subtle disc edema in patients felt to not have papilledema.

Added:

  • Future studies using OCT imaging of the optic nerve head could help evaluate this question.

  1. Why is papilledema not a university clinical feature of raised ICP?:

Discuss that unilateral papilledema can be present and that 10% can have >=2 grade papilledema difference so IIHWOP can exist.

Added:

  • Anatomical anomalies within the optic nerve sheath have been postulated which might prevent the development of papilloedema [12][62] Unilateral papilledema can be present in IIH [63] and a portion of patients have a 2 grade difference in their papilledema (as measured using the Frisen grading system), and this may be considered to be the reason why papilledema has yet to develop in some with IIHWOP.
  1. Management of IIHWOP

Caution that medications and especially surgical interventions are not necessarily benign so caution must be taken to not be overly aggressive for IIHWOP since there is no threat to the vision.

Added:

  • Unlike IIH there does not appear to be a threat of visual loss in IIHWOP [62], and hence why caution in treatments that are invasive or have significant adverse profiles should be avoided.

Reviewer 2 Report

The major considerations as follows:

  1. In Introduction: “…found in women of working age,” should be “… found in women of fertile age,”.
  2. In Section 4: “Progress is being made in non-invasive ways of measuring ICP in those with IIH including optical coherence tomography…” is not correct. OCT can not measure ICP, but measure the thickness of retinal nerve fibers, which may reflect papilloedema.
  3. In section 5: Please address which factors that may affect OP by lumbar puncture (age, body weight, stress, spinal needle seize, the lumbar measurement system, etc).
  4. In section 6: authors refer to agreement about diagnosis of IIHWOP in the UK that LP OP should not be taken in isolation but should be considered within the clinical context. The question is: can one make the diagnosis IIH in case a patient has typical clinical and neuroimaging features of IIH, but with normal LP OP?
  5. In section 7: Telemetric ICP may lead to identify more IIHWOP with bilateral TSS in patients with CDH. Shall this group of patients undergo stenting?
  6. In section 8: According to the data in Table 1, can one consider bilateral transverse sinus stenosis as biomarker of IIH? Regarding spontaneous venous pulsations, what is the best method to detect them?
  7. In section 12: Papilloedema is caused by the disturbances of axonal transport in the optic nerves due to intracranial hypertension. How can authors explain the relationship between axonal transport and CSF flow in ONS-SAS-IC-SAS.
  8. In section 13. Do authors recommend medical management with acetazolamide or topiramate IIHWOP based on the pathophysiology of IIHWOP?

Author Response

Reviewer 2

We would like to thank the reviewer for their time and helpful comments.

The major considerations as follows:

  1. In Introduction: “…found in women of working age,” should be “… found in women of fertile age,”.

Corrected to ‘child bearing age’.

  1. In Section 4: “Progress is being made in non-invasive ways of measuring ICP in those with IIH including optical coherence tomography…” is not correct. OCT can not measure ICP, but measure the thickness of retinal nerve fibers, which may reflect papilloedema.

Corrected to Progress is being made in non-invasive ways to act as a surrogate measure of ICP in those with IIH include optical coherence tomography[25][26][27] and ultrasound and are further discussed in section 7.0.

  1. In section 5: Please address which factors that may affect OP by lumbar puncture (age, body weight, stress, spinal needle seize, the lumbar measurement system, etc).

We have corrected this to include:

A recent large population-based study showed that LP OP can vary significantly among individuals. Higher OPs were associated with higher BMI and younger age. It reported that 15.8% of normal individuals had an OP ≥ 200 mmH20 and 2% had an OP ≥ 200 mmH20. In addition, they reported a group of 79 patients who had two LPs within 2.5 years and in 12.7% there was a difference of ≥50 mmH2O [29].

  1. In section 6: authors refer to agreement about diagnosis of IIHWOP in the UK that LP OP should not be taken in isolation but should be considered within the clinical context. The question is: can one make the diagnosis IIH in case a patient has typical clinical and neuroimaging features of IIH, but with normal LP OP?

Well that is the question! A number of factors. Was the LP OP reliable? Etc!

  1. In section 7: Telemetric ICP may lead to identify more IIHWOP with bilateral TSS in patients with CDH. Shall this group of patients undergo stenting?

I am sure we will see studies on stenting in CDH. 

  1. In section 8: According to the data in Table 1, can one consider bilateral transverse sinus stenosis as biomarker of IIH?

We have referenced Mallery et al. where 3 out 4 MRI findings:

This has been recently validated. A combination of any 3 of 4 MRI features was found to be nearly 100% specific, with a sensitivity of 64%, for a diagnosis of IIH without papilloedema in patients with chronic headache, no papilloedema and elevated lumbar puncture opening pressure.  

  1. Regarding spontaneous venous pulsations, what is the best method to detect them?

Added: Spontaneous retinal venous pulsation (SVP) is a subtle variation in the calibre of the retinal vein(s) as they cross the optic disc [45] 10% of the normal population do not have detectable SVP [46][47]. SVP can be detected clinically on examination, and have been recorded using video ophthalmoscopy and optical coherence tomography (OCT) imaging.

To my knowledge and searches I cannot find a head to head study, to guide what is the best way to detect them.   

12: Papilloedema is caused by the disturbances of axonal transport in the optic nerves due to intracranial hypertension. How can authors explain the relationship between axonal transport and CSF flow in ONS-SAS-IC-SAS.

Papilloedema is thought to be due to increased ICP that is transmitted to the SAS of the optic nerve where it causes stasis of axonal transport due to mechanical pressure. However, in IIHWOP papilloedema does not occur. The work of Killer et al. suggests that the ONS-SAS is not continuous with the intracranial SAS, and given that the ONS-SAS has numerous trabeculae and septae that reduce both CSF volume and CSF flow I think it is challenge to explain the exact relationship.

  1. In section 13. Do authors recommend medical management with acetazolamide or topiramate IIHWOP based on the pathophysiology of IIHWOP?

This section details : The majority of case series indicated that headaches improved following medical treatment with acetazolamide and topiramate, and intervention by repeated CSF drainage by LP.